# Past speculations of future health technologies: a description of technologies predicted in 15 forecasting studies published between 1986 and 2010

Lucy Doos,[1] Claire Packer,[1] Derek Ward,[1] Sue Simpson,[1] Andrew Stevens[2]

## ABSTRACT

**Objective** To describe and classify health technologies predicted in forecasting studies.

**Design and methods** A portrait describing health technologies predicted in 15 forecasting studies published between 1986 and 2010 that were identified in a previous systematic review. Health technologies are classified according to their type, purpose and clinical use; relating these to the original purpose and timing of the forecasting studies.

**Data sources** All health-related technologies predicted in 15 forecasting studies identified in a previously published systematic review.

**Main outcome measure** Outcomes related to (1) each forecasting study including country, year, intention and forecasting methods used and (2) the predicted technologies including technology type, purpose, targeted clinical area and forecast timeframe.

**Results** Of the 896 identified health-related technologies, 685 (76.5%) were health technologies with an explicit or implied health application and included in our study. Of these, 19.1% were diagnostic or imaging tests, 14.3% devices or biomaterials, 12.6% information technology systems, eHealth or mHealth and 12% drugs. The majority of the technologies were intended to treat or manage disease (38.1%) or diagnose or monitor disease (26.1%). The most frequent targeted clinical areas were infectious diseases followed by cancer, circulatory and nervous system disorders. The most frequent technology types were for: infectious diseases—prophylactic vaccines (45.8%), cancer—drugs (40%), circulatory disease—devices and biomaterials (26.3%), and diseases of the nervous system—equally devices and biomaterials (25%) and regenerative medicine (25%). The mean timeframe for forecasting was 11.6 years (range 0–33 years, median=10, SD=6.6). The forecasting timeframe significantly differed by technology type (p=0.002), the intent of the forecasting group (p<0.001) and the methods used (p<001).

**Conclusion** While description and classification of predicted health-related technologies is crucial in preparing healthcare systems for adopting new innovations, further work is needed to test the accuracy of predictions made.

► The study summarises the forecast technologies by type, for example, medical devices, diagnostic tests or drugs; by clinical purpose, for example, prevention, diagnosis or treatment and by targeted clinical area by International Classification of Diseases, 10th revision chapter and relates this to the forecasting purpose and timeframe.

► The analysis and the health-related technologies identified are constrained by the methods used in the forecasting studies, for example, by the areas of interest of the experts involved.

► Although the study highlights the identified technologies, the retrospective accuracy of the studies remains to be researched.

## INTRODUCTION

People are in general living longer and surviving previously life-threatening illnesses. A proportion of this effect is attributed to the use of novel health interventions and technologies, such as new vaccines to prevent disease; new diagnostic tests to diagnose diseases earlier and with improved accuracy and new drugs, devices and surgical procedures to treat diseases more effectively. In addition, new health technologies may also be used to enhance function and improve quality of life, such as prostheses, sensory aids and cosmetic enhancements. Technological advances and innovation are leading to new health interventions becoming available to healthcare markets at an increasing speed; these often cost more than current alternatives and significantly affect the cost of healthcare services and delivery.[1 2] To ensure that patients can access the most effective and

[1]NIHR Horizon Scanning Research and Intelligence Centre, Institute of Applied Health Research, University of Birmingham, Birmingham, UK
[2]Institute of Applied Health Research, University of Birmingham, Birmingham, UK

**Correspondence to**
Dr Lucy Doos;
l.doos@bham.ac.uk

cost-effective interventions, and because many health systems work within constrained budgets, healthcare systems need to be prepared to respond to new developments. A crucial step in such preparation is the accurate identification and forecasting of likely significant technological healthcare developments.[3][4]

Forecasting is widely used for planning and strategic decision-making around industrial and economic development at organisational, regional and national levels.[5] Within healthcare, forecasting can be used to build strategies for supporting innovators and companies to develop new interventions, to plan future research programmes and delivery of healthcare and to prepare health services and personnel to respond to technological developments of benefit to patients and service provision. Health technology forecasting uses various methods, such as Delphi surveys and reviews of the literature[6–8] to identify emerging health technologies that are intended to address important unmet healthcare needs or that need additional evaluation and to analyse their potential impact on patients and health systems.[4]

To get the maximum benefit from forecasting, it is important that the methods used to identify emerging interventions and technologies are appropriate for the aims and timeframe of the forecasting exercise and also as accurate as possible.

In a recent systematic review,[9] we found 15 forecasting studies with 3 to 20 year timeframes that identified emerging health-related topics and technologies predicted between 1986 and 2010 from 12 high-income countries, including 6 from the UK. Identifying and classifying the predicted new technologies into meaningful groupings is

important for healthcare system preparedness, allowing planned assessment and adoption as appropriate. Here we describe and classify the health technologies predicted in the forecasting studies according to their type, purpose and clinical use. We relate these to the forecast time period and develop an overview of the technological and clinical frontiers of innovation in health and healthcare provision.

## METHODS

See table 1 for a full list of the 15 studies identified in our prior systematic review. We extracted all forecast topics from the 15 forecasting studies using a standardised set of data for each of these predicted topics and on the source study. Data related to the forecast topics included the name of the topic as written in the forecasting reports and papers, and the predicted forecast timeframe. Data related to the study included the country or region of the forecast, year the study was undertaken, its intention and remit, the study forecasting timeframe and the number and type of forecasting methods used.

### Identification and inclusion of health technologies

For final inclusion and analysis, topics had to be health technologies as defined by the National Institute of Health Research (NIHR) Health Technology Assessment (HTA) programme[10]: 'any method used to promote health; prevent and treat disease; and improve rehabilitation or long-term care. They are not confined to new drugs and include any intervention used in the treatment, prevention or diagnosis of disease'. We also included topics that

**Table 1** Fifteen forecasting studies identified in the prior systematic review

| Study name or first author | Year of forecast | Country |
|---|---|---|
| Dutch Steering Committee on Future Health Scenarios (STG), 1988 | 1986 | The Netherlands |
| Spiby, 1988 | 1988 | UK |
| Loveridge et al[6] | 1994 | UK |
| Stevens et al[7] | 1995 | UK |
| Karim, National Research and Technology Foresight Project, 1999 | 1996 | South Africa |
| Operating theatre of the year, 2010; Department of Trade and Industry report, 1999 | 1996 | UK |
| Cahill and Scapolo[18] | 1998 | Europe |
| Daar et al | 2002 | Developing countries |
| Technology foresight towards 2020_China | 2003 | China |
| British Telecommunications (BT) calendar | 1997 & 2005 | UK |
| Tremblay and Yiu | 2006 | Canada |
| Food and Drug Administration surveys | 1998 and 2008 | USA |
| Institute of the Future, 2009 | 2009 | USA |
| Science & Technology Foresight Survey, 2010 | Every 5 years from 1971, latest survey 2009–2010 | Japan |
| UK technology and innovation futures for the 2020s, 2010 | 2010 | UK |

*Source*: Doos et al.[9]

related to the recognition of, or change in, health states or emotions, or health-related behaviours. All topics needed to have an explicit or implied health-related application or identified patient group for final inclusion. Topics were excluded if they related exclusively to foods, plants, animals, insects, new sources of fuel or energy, environmental contamination, legislation or health insurance.

LD read all the included studies and manually extracted all technologies mentioned in the forecast from the published text and related tables. Two authors (CP and LD) independently scrutinised each identified topic and applied the inclusion and exclusion criteria. Where there was insufficient information on a topic to make the inclusion/exclusion decision, we undertook brief and very limited searches on the internet to find additional information.

All topics classed as a health technology with an explicit or implied health-related application were subsequently coded based on their technology type, intended purpose and targeted clinical use.

### Technology type

we used broad categories: assistive devices and rehabilitation aids, devices and biomaterials, diagnostic tests and imaging, drugs, information technology (IT) systems, electronic health systems (eHealth) and mobile health (mHealth), medical equipment, non-surgical therapy, organisational programmes, prophylactic vaccines, population programmes, regenerative medicine approaches and therapeutic procedures.

### Technology purpose

we used categories that relate to the intended point on the patient pathway: health promotion and the prevention of disease, diagnosis and monitoring of disease, treatment and management of symptoms and disease, and rehabilitation. We supplemented these with two groupings that categorised technologies that support the provision of care (1) supporting patients receiving care and designed to be used by individuals (patients, caregivers or healthcare professionals) and (2) supporting the provision of care and/or increasing service efficiency.

### Targeted clinical area (specialty)

we used the international statistical classification of diseases and related health problems 10th revision (ICD-10)[11] chapter headings relating to diseases and other morbid conditions (chapters I–XVII); symptoms, signs and abnormal findings (chapter XVIII); injuries, poisoning and other consequences of external causes (chapter XIX); external causes of diseases and morbid conditions chapter XX) and factors influencing health status and contact with health services for people not currently sick (chapter XXI). Chapter XXI includes contraception, technologies used in population screening, health promotion and disease prevention, and general rehabilitation.

All topics with disputed inclusion, exclusion or categorisation were initially discussed between two authors (LD and CP) and disagreements resolved where possible.

Where disagreements remained, topics were discussed with the other authors as a group and the final categorisation agreed through consensus.

### Technology forecast timeframe

For each health-related technology forecast, we calculated the difference in years between each technology's predicted year of impact as identified in the original study, and the year when the prediction was made, that is, the year of the original study. Technology forecasts were classified as short term if their forecasting timeframe was less than 3 years, short to medium term if it was 3–10 years, medium to long term if it was 11–20 years and long term for those with timeframes of more than 20 years. Although the original systematic review excluded forecasting studies which exclusively adopted a short timeframe of less than 3 years, some studies with longer timeframes also provided predictions with shorter timeframes, and these were included in the analysis.

### Data analysis

We analysed the data using IBM SPSS statistics (V.22) for Windows. We present descriptive analyses as means and SD for normally distributed continuous variables. Statistically significant differences were determined using ANOVA for continuous normally distributed data and $X^2$ for dichotomous variables.

## RESULTS

We identified 896 predicted topics from the 15 forecasting studies. Of these, we judged that 685 (76.5%) were health-related technologies with an explicit or implied health-related application. The most frequently excluded topics were those that described advances in the underpinning scientific knowledge (8.7%), such as determination of whole human DNA base sequence and identifying genetic links to diseases. A full list of the included health-related technologies is attached as an online supplementary appendix 1.

Of the 685 health-related technologies, 52.3% were forecast from six studies by governmental organisations (such as the UK Department of Trade and Industry[12]), 21.9% by commercial/consultancy organisations (such as British Telecommunications), 14.5% by policy planning groups (such as Loveridge et al,[6] 8.9% by research groups (such as those by Stevens et al[7] and 2.5% by non-profit organisations (such as Institute of the Future[13]). Looking at the purpose of forecasting, our data showed that two-thirds (68%) of the health technologies were forecast for policy planning purposes, 18.1% for research purposes and 13.9% for commercial purposes.

Nearly half (48.9%) of the identified technologies were from six UK-based studies and reports, 15% were from Japan, 11.7% from the Netherlands, 11.4% from the USA and 8% from Canada. The remainder were from a homogenous group of developing countries and South Africa (3.8%), other European countries (0.9%) and China (0.3%).

**Table 2** Health technologies by technology type

| Technology type | n | % |
|---|---|---|
| Diagnostic tests and imaging | 131 | 19.1 |
| Devices and biomaterials | 98 | 14.3 |
| IT systems, eHealth and mHealth | 86 | 12.6 |
| Drugs (not prophylactic vaccines) | 82 | 12.0 |
| Regenerative medicine | 80 | 11.7 |
| Medical equipment | 37 | 5.4 |
| Therapeutic procedures | 36 | 5.3 |
| Unknown | 33 | 4.8 |
| Organisational programmes | 26 | 3.8 |
| Prophylactic vaccines | 24 | 3.5 |
| Population programmes | 24 | 3.5 |
| Non-surgical therapy | 15 | 2.2 |
| Other | 9 | 1.3 |
| Assistive devices | 4 | 0.6 |
| Total | 685 | 100 |

IT, information technology; eHealth, electronic health; mHealth, mobile health.

## Technology type

Of the 685 health technologies forecast, 19.1% were diagnostic tests and imaging technologies, such as advanced ultrasound imaging systems and molecular diagnosis; 14.3% were devices and biomaterials, such as tissue engineered devices and drug impregnated devices; 12.6% were IT systems, eHealth and mHealth, such as electronic prescriptions and telemedicine; 12% were drugs, such as new anaesthetic vapours and electronically activated drugs and 11.7% were regenerative medicine approaches, such as gene therapy for diabetes and widespread use of gene therapy for familial hypercholesterolaemia (table 2 and online supplementary appendix 1). Drug technologies were more commonly forecast in studies carried out by researchers (36.1%) than other groups, while technologies for diagnostic tests and imaging were more commonly forecast in studies by governmental agencies than other groups (table 3). Regenerative medicine approaches were the most commonly forecast technologies (23.2%) by studies with a commercial intent while diagnostic tests and imaging were the most commonly forecast technologies by studies with policy planning and research intentions (20.6% and 23.4%, respectively) (p<0.001).

## Technology purpose

Of the 685 health technologies included in the forecasting reports, 38.1% were for the treatment and management of symptoms and disease, such as development of effective treatment for amyotrophic lateral sclerosis and the practical use of gene therapy for genetic disorders; 21.6% for the diagnosis and monitoring of disease, for example, a diagnostic biochip for cancer and the further development of three-dimensional imaging techniques; 13% for

health promotion and the prevention of disease, such as the first effective vaccine of HIV and DNA vaccines for AIDS, malaria, hepatitis B and certain cancers; 10.2% for supporting the provision of care, for example, telemonitoring and teleconsultation; 8% for supporting patients receiving care, such as smart pill bottles to remotely monitor medication use and thought-controlled robots for personal healthcare; and 4.7% for rehabilitation, for example, artificial legs and robotic prosthetics.

Technologies for treatment and disease management were the most common regardless of the intention of the study: 35.8% among studies with a commercial intent, 35.8% for policy planning and 48.4% for research. Technologies for treatment and disease management were also the most common technologies forecast by all the various groups (60.7% among researchers, 39.9% among governmental organisations and 35.8% among commercial organisations).

## Targeted clinical area

Using the ICD-10 codes to define the targeted clinical area, just over half of the health technologies (50.4%) were for diseases and conditions that were not specified or that could not be coded or that crossed multiple specialities. Of the 340 health technologies that could be coded to a specific ICD-10 chapter, the top five targeted clinical areas were the following: (1) infectious and parasitic diseases ICD-I (14.1%), such as development of vaccines for AIDS and vaccines for malaria, (2) neoplasms ICD-II (11.8%), such as expanding metal stents for oesophageal cancer and diagnostic biochip for cancer, (3) diseases of the circulatory system ICD-IX (11.2%), such as artificial muscles in replace of heart transplants and implantable vascular stents, (4) diseases of the nervous system ICD-VI (10.6%), such as artificial brain cells and artificial peripheral nerves and (5) factors influencing health status and contact with health services ICD-XXI (10.3%), such as a personal wearable health monitor and extension of average lifespan to over 100.

Congenital malformations, deformations and chromosomal abnormalities ICD-XVII and symptoms, signs and abnormal clinical and laboratory findings, not elsewhere classified ICD-XVIII had the fewest forecast technologies (0.3% each).

Table 4 shows the technology types and purposes for the forecast technologies within the five most frequent targeted clinical areas. The most frequent technology type forecast for infectious and parasitic diseases were prophylactic vaccines (45.8%); for cancer were drugs (40%); for diseases of the circulatory system were devices and biomaterials (26.3%); for diseases of the nervous system were devices and biomaterials (25%) and regenerative medicine (25%) and for factors influencing health status and contact with health services including screening and rehabilitation, devices and biomaterials (31.4%). The most frequent technology purpose was health promotion and disease prevention for both infectious and parasitic diseases (ICD-I) (60.4%) and factors influencing health

**Table 3** Technology type by forecasting group

| Technology type | Commercial organisation | Consultancy agency | Governmental agency | Independent non-profit organisation | Researchers | Policy research group | Total |
|---|---|---|---|---|---|---|---|
| Assistive devices | 3 (3.2%) | 0 (0.0%) | 1 (0.3%) | 0 (0.0%) | 0 (0.0%) | 0 (0.0%) | 4 (0.6%) |
| Devices and biomaterials | 16 (16.8%) | 6 (10.9%) | 56 (15.6%) | 3 (17.6%) | 7 (11.5%) | 10 (10.1) | 98 (14.3%) |
| Diagnostic tests and imaging | 6 (6.3%) | 26 (47.3%) | 63 (17.6%) | 3 (17.6%) | 9 (14.8%) | 24 (24.2%) | 131 (19.1) |
| Drugs (non- prophylactic vaccine) | 7 (7.4%) | 2 (3.6%) | 31 (8.7%) | 1 (5.9%) | 22 (36.1%) | 19 (19.2%) | 82 (12.0%) |
| IT systems, eHealth, mHealth | 18 (18.9%) | 7 (12.7%) | 43 (12.0%) | 3 (17.6%) | 3 (4.9%) | 12 (12.1%) | 86 (12.6%) |
| Medical equipment | 2 (2.1%) | 5 (5.5%) | 31 (8.7%) | 0 (0.0%) | 1 (1.6%) | 0 (0.0%) | 37 (5.4%) |
| Non-surgical therapy | 1 (1.1%) | 0 (0.0%) | 9 (2.5%) | 1 (5.9%) | 3 (4.9%) | 1 (1.0%) | 15 (2.2%) |
| Population programme | 5 (5.3%) | 0 (0.0%) | 9 (2.5%) | 3 (17.6%) | 1 (1.6%) | 6 (6.1%) | 24 (3.5%) |
| Prophylactic vaccine | 0 (0.0%) | 0 (0.0%) | 18 (5.0%) | 0 (0.0%) | 0 (0.0%) | 6 (6.1%) | 24 (3.5%) |
| Regenerative medicine | 22 (23.2%) | 1 (1.8%) | 48 (13.4%) | 1 (5.9%) | 4 (6.6%) | 4 (4.0%) | 80 (11.7%) |
| Therapeutic procedure | 0 (0.0%) | 1 (1.8%) | 20 (5.6%) | 0 (0.0%) | 6 (9.8%) | 9 (9.1%) | 36 (5.3%) |
| Organisational programme | 2 (2.1%) | 10 (10.6%) | 11 (3.1%) | 1 (5.9%) | 4 (6.6%) | 2 (2.0%) | 26 (3.8%) |
| Unknown | 9 (9.5%) | 2 (3.6%) | 16 (4.5%) | 1 (5.9%) | 1 (1.6%) | 4 (4.0%) | 33 (4.8%) |
| Others | 4 (4.2%) | 1 (1.8%) | 2 (0.6%) | 0 (0.0%) | 0 (0.0%) | 2 (2.0%) | 9 (1.3%) |
| Total | 95 (100%) | 55 (100%) | 358 (100%) | 17 (100%) | 61 (100%) | 99 (100%) | 685 (100%) |

IT, information technology; eHealth, electronic health; mHealth, mobile health.

**Table 4** Technology type and purpose for the five most common ICD-10 chapter headings

| | Infectious disease ICD-I | Cancer ICD-II | Circulatory disorders ICD-IX | Nervous system ICD-VI | Health status, screening, rehab ICD-XXI | Total |
|---|---|---|---|---|---|---|
| **Technology type** | | | | | | |
| Assistive devices | – | – | – | – | 3 (8.6) | 3 (1.5) |
| Diagnostic tests and imaging | 5 (10.4) | 12 (30.0) | 7 (18.4) | 6 (16.7) | 2 (5.7) | 32 (16.2) |
| Devices and biomaterials | 3 (6.3) | 1 (2.5) | 10 (26.3) | 9 (25.0) | 11 (31.4) | 34 (17.3) |
| IT systems, eHealth and mHealth | – | – | – | 1 (2.8) | 3 (8.6) | 4 (2.0) |
| Drugs (not prophylactic vaccines) | 10 (20.8) | 16 (40.0) | 7 (18.4) | 2 (5.6) | 5 (14.3) | 40 (20.3) |
| Regenerative medicine | – | 2 (5.0) | 7 (18.4) | 9 (25.0) | – | 18 (9.1) |
| Medical equipment | 2 (4.2) | – | – | 1 (2.8) | 1 (2.9) | 4 (2.0) |
| Therapeutic procedures | 1 (2.1) | 1 (2.5) | 3 (7.9) | 4 (11.1) | – | 9 (4.6) |
| Organisational programmes | 3 (6.3) | – | – | – | 1 (2.9) | 4 (2.0) |
| Prophylactic vaccines | 22 (45.8) | – | – | – | 1 (2.9) | 23 (11.7) |
| Population programmes | 1 (2.1) | 2 (5.0) | 2 (5.3) | 1 (2.8) | 3 (8.6) | 9 (4.6) |
| Non-surgical therapy | – | 2 (5.0) | – | 1 (2.8) | – | 3 (1.5) |
| Other/unknown | 1 (2.1) | 4 (10.0) | 2 (5.3) | 2 (5.6) | 5 (14.3) | 14 (7.1) |
| Total | 48 (100%) | 40 (100%) | 38 (100%) | 36 (100%) | 35 (100%) | 197 (100%) |
| **Technology purpose** | | | | | | |
| Diagnosis and monitoring of disease | 7 (14.6) | 12 (30.0) | 8 (21.1) | 7 (19.4) | 1 (2.9) | 35 (17.8) |
| Health promotion and disease prevention | 29 (60.4) | 5 (12.5) | 5 (13.2) | 1 (2.8) | 18 (51.4) | 58 (29.4) |
| Treatment and disease management | 11 (22.9) | 22 (55.0) | 25 (65.8) | 22 (61.1) | 1 (2.9) | 81 (41.1) |
| Rehabilitation | 0 (0.0) | 0 (0.0) | 0 (0.0) | 4 (11.1) | 15 (42.9) | 19 (9.6) |
| Other and unknown | 1 (2.1) | 1 (2.5) | 0 (0.0) | 2 (5.6) | 0 (0.0) | 4 (2.0) |
| Total | 48 (100%) | 40 (100%) | 38 (100%) | 36 (100%) | 35 (100%) | 197 (100%) |

ICD-10, International Classification of Diseases, 10th revision; IT, information technology; eHealth, electronic health; mHealth, mobile health.

**Table 5** Mean technology prediction timeframe (years) by technology type*

| Technology type | n | Mean timeframe, years (SD) |
|---|---|---|
| Non-surgical therapy | 14 | 9.07 (5.7) |
| Diagnostic tests and imaging | 113 | 10.11 (6.0) |
| Drugs (not prophylactic vaccines) | 75 | 10.31 (6.3) |
| Therapeutic procedures | 31 | 10.45 (8.2) |
| Organisational programmes | 25 | 10.48 (5.3) |
| IT systems, eHealth and mHealth | 77 | 11.08 (5.5) |
| Devices and biomaterials | 88 | 11.93 (6.8) |
| Assistive devices | 3 | 12.33 (17.9) |
| Other/unknown | 40 | 13.35 (7.2) |
| Medical equipment | 36 | 13.53 (2.7) |
| Regenerative medicine | 73 | 13.58 (7.4) |
| Prophylactic vaccines | 13 | 14.00 (6.3) |
| Population programmes | 19 | 14.53 (5.9) |
| Total | 607 | 11.59 (6.6) |

*p<0.002.
IT, information technology; eHealth, electronic health; mHealth, mobile health.

status and contact with health services including screening and rehabilitation (ICD-XXI) (51.4%). Technologies for treatment and disease management were most common for cancer (55.0%), diseases of the circulatory system (65.8%) and nervous system disorders (61.1%).

### Technology forecast timeframe

Data were available to calculate the prediction timeframe for 607 of the identified health technologies. The mean timeframe for forecasting predictions was 11.6 years (range 0–33 years, median=10, SD=6.6). There was a significant difference in the mean prediction timeframe by technology type (p=0.002), with non-surgical therapies having the shortest mean forecast timeframe (9.07 years) and population health programmes having the longest mean forecast (14.53 years) (table 5). There were no statistically significant differences in the mean duration of the technology prediction timeframe by the technology purpose and we were not able to detect any particular trend in the technology types or purposes of forecast overtime.

There was a statistically significant difference in the forecast timeframe by the type of group undertaking the forecasting (p<0.001): 58% of the long-term forecasts (>20 years) were made by policy planning groups and 51% of short-term forecasts (<3 years) were by research groups.

Technologies forecast by studies with research intentions had the shortest mean forecast timeframe (4.49 years) while those for policy-making had the longest timeframe (14.16 years) (p<0.001). Similarly, technologies forecast by researchers had significantly shorter mean timeframes (3.56 years) compared with those made by governmental agencies which had the longest mean forecast timeframes (13.42 years) (p<0.001).

There was a statistically significant difference in the mean forecast timeframe when considering the forecasting method (p<0.001) (figure 1). Technologies forecast using methods that included creativity-based methods which require brainstorming, such as scenario building, had a significantly longer forecasting timeframe (13.5 years) compared with those did not include creativity-based methods such as literature reviews (9.7 years)

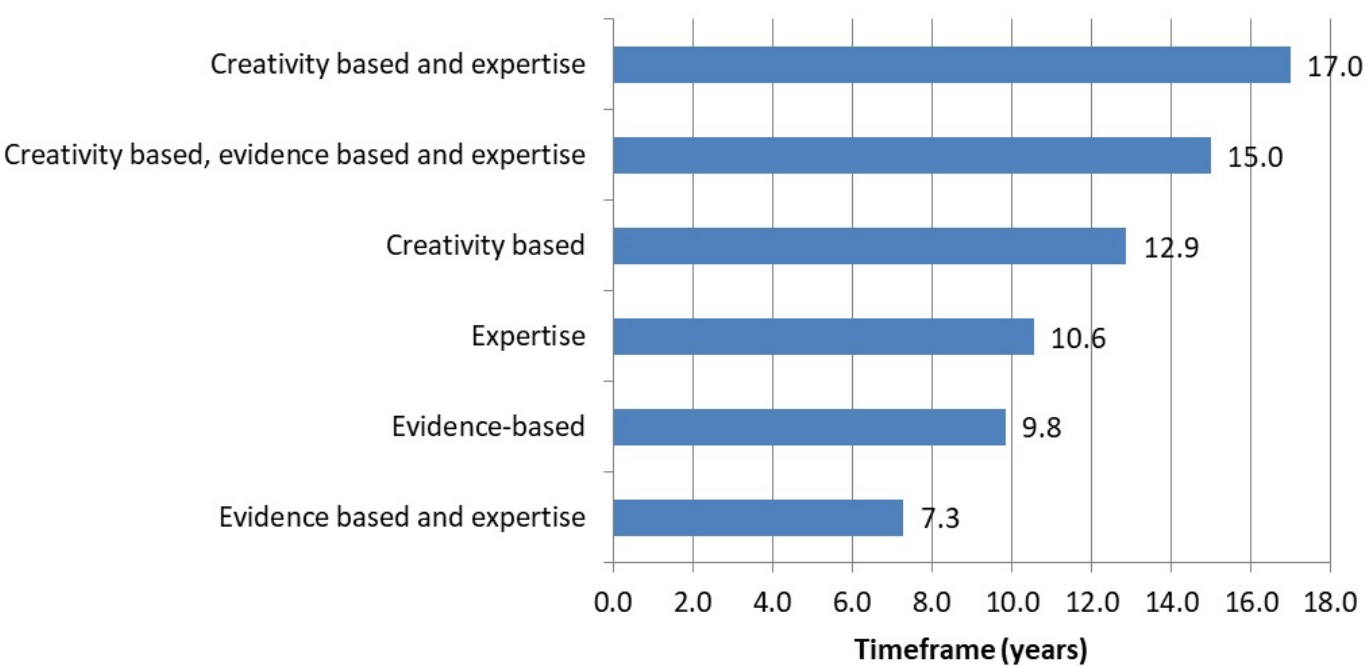

**Figure 1** Mean forecast timeframe by methods of forecasting.*p<0.001

(p<0.001). On the other hand technologies forecast using methods that included evidence-based methods such as literature reviews and evidence from HTA organisations, had significantly shorter forecasting timeframes (8.2 years) than those that did not include evidence-based methods (10.8 years) (p<0.001).

## DISCUSSION

Being aware of forthcoming developments in healthcare and preparing for the future is desirable for health policy makers, whether developing healthcare policy, directing future research or commissioning health services. The use that health policy makers make of forecasting depends on having systems in place that receive and act on the information at an appropriate time. There are few examples of such systems in the published literature[14]; however, the relationship between the UK's National Institute for Health and Care Excellence and NIHR Horizon Scanning Research and Intelligence Centre does provide a model of how shorter-term early awareness intelligence can shape prioritisation decisions.[15] We found that more than two-thirds of the identified technologies were forecast for policy planning purposes and over half were forecast by governmental organisations, and that the majority of long-term forecasts were made by policy planning groups as opposed to research groups. This finding supports the premise that the majority of forecasting is carried out for service preparedness and long-term planning and strategic decision-making.[5]

Our study identified and classified a number of important health-related technologies, which health services may have already had to manage or will have to do so in the future. Our finding that nearly 60% of the technologies were either for the treatment and management of symptoms and disease or for the diagnosis and monitoring of disease can perhaps be explained by the methods used for forecasting. Many of the studies involved soliciting the opinion of experts and, although it is not possible to be certain about the areas of interest of those involved, their expertise may have made them more likely to focus on diagnosis and treatment than, for example, organisational programmes or rehabilitation. Regenerative medicine approaches were the most commonly forecast technologies by studies with a commercial intent. In contrast, drugs and related technologies (but not prophylactic vaccines) were the most common forecasts made by researchers.

The four clinical areas with the greatest number of forecast health technologies: infectious diseases, cancer, circulatory and nervous system diseases and disorders, include many of the major causes of death and disability in countries today. Again we are not able to determine if this merely reflects the interests of the groups carrying out the forecasting and their areas of expertise or whether it truly represents the technology innovation frontier. It is understandable that the forecasts for infectious disease included many prophylactic vaccines as these have proven to be a cost-effective preventive strategy for their prevention.[16] This was also supported by our finding that forecasts of vaccine related technologies were only made by governmental agencies and policy research groups. In cancer, forecasting included both drugs and diagnostic tests and imaging, perhaps reflecting increasing targeting in novel cancer therapies.[17 18] Several forecasts of regenerative medicine approaches for nervous system diseases and disorders is also interesting, with many of these diseases not currently curable or effectively treated.[19 20]

Our finding that forecasts which included creativity-based methods, such as scenario building and analysis had, on the whole, longer forecast timeframes than studies that included evidence-based methods may have a rationale. Evidence-based methods require there to be research evidence on technologies, which will therefore probably be nearer to any predicted impact. In contrast, creativity-based methods require experts to think more widely about what the future could be and are not therefore constrained by what is actually being currently researched and published. This may also explain our finding that just over half of the short-term forecasts were made by research groups who may be more likely to use published evidence to inform their deliberations.

We believe that this is the first study to comprehensively identify and summarise health-related technologies predicted in past forecasting projects and to categorise and describe them. Although as many predicted topics as possible were included, we did exclude a substantial number. This was often because we could not envisage a future health application or because there was insufficient information to make a judgement (and many topics had very sparse titles or descriptive information). By excluding topics in this way, we may have excluded some that would have had an explicit health application within a reasonable timeframe. In addition, it is likely that we were unable to identify a number of relevant forecasts made by commercial organisations, which will have been conducted and disseminated in confidence.

Our description and classification of predicted health-related technologies from prior forecasting studies provides an overview of the technological and clinical frontiers of innovation in health and healthcare provision. To complete our evaluation of the accuracy of previous forecasting predictions and the forecasting methods used, the accuracy of the predictions made should also be assessed. This may be difficult across all technology types and clinical areas, but possible by limiting the evaluation to specific clinical areas and/or technology types or to those topics with more obvious health applications and timeframes that have already occurred.

**Contributors** All authors have directly participated in the planning and execution of the study. LD extracted the study data. LD and CP independently reviewed the technologies to be included. LD and CP drafted the paper and DW, SS and AS critically revised the manuscript. All authors agreed the final version of the paper. All authors had full access to all of the data.

**Funding**  At the time of the study, LD, CP, DW and SS were funded by the UK's NIHR and AS was partly funded by the UK's NIHR. This report presents independent research funded by the NIHR.

**Disclaimer**  The views expressed in this publication are those of the authors and not necessarily those of the National Health Service, the NIHR or the Department of Health.

**Competing interests**  None declared.

**Provenance and peer review**  Not commissioned; externally peer reviewed.

**Data sharing statement**  A full list of health technologies included in the study is available in a supplemental document submitted with this article. More details on data are available from the corresponding author at l.doos@bham.ac.uk.

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
