## [Reviewer comments · BMJ Open]

ARTICLE DETAILS

TITLE (PROVISIONAL)	Past speculations of future health technologies: a description of technologies predicted in 15 forecasting studies 1986-2010
AUTHORS	Doos, Lucy; Packer, Claire; Ward, Derek; Simpson, Sue; Stevens, Andrew

VERSION 1 - REVIEW

REVIEWER	Jeffrey Lerner ECRI Institute, USA My institute provides a forecasting service, although it is hard for me to see this as competing with this article, as opposed to being a qualification for reviewing it. So this is a disclosure. Also, an article I published in Health Affairs is referenced in the article.
REVIEW RETURNED	08-Feb-2017

GENERAL COMMENTS	1. The article is interesting for policy makers as well as horizon scanners “in the weeds” doing the daily work of horizon scanning, though I suspect most interesting to the latter.2. It illustrates some of the differences that emerge from horizon scanning done for different purposes and by different kinds of entities (e.g., commercial research, policy, governmental), though I believe readers would benefit from additional discussion of the differences. One naturally suspects that a great deal of commercial horizon scanning is not represented in the data sources, which may be worth acknowledging.3. I believe the introduction could include a sentence stating that some technologies are also used to enhance function, such as cosmetic enhancements, rather than just to prevent, diagnose or treat patients.4. The data analysis provides an interesting tool to contrast and compare to the national horizon scanning activities conducted in the U.S. from 2010 through 2016 and those conducted in the EU. Looking at this analysis and U.S.-based analyses of the AHRQ horizon scanning activity previously published to the AHRQ website, one can discern both similarities and differences in priority setting, timeframes used for scanning and forecasting, target clinical areas of interest, and technology types. The AHRQ system was halted December 31, 2015, but until then was the largest system in the world, I believe. It may be worth a mention.5. The broad categories used for classifying technology type are less informative than more granularly defined categories. For example, there may be interesting trends to observe in the “drugs” category—such as “targeted therapies” or “biologics”—but that would require more granular analysis and categorization that may not be possible from the studies from which data were abstracted.
--

REVIEWER	Mattijs Lambooj National Institute for Public Health and the Environment (RIVM). The Netherlands
REVIEW RETURNED	28-Feb-2017

GENERAL COMMENTS	The paper "Past speculations of future health technologies: what did they predict?" presents an inventory on forecast topics conducted in the previous decades and describes the health technologies, technology purpose, targeted clinical area and forecast time frame. Abstract: The objective is to describe and classify a number of health technologies predicted in forecasting studies. Maybe the format of the journal does not allow for presenting an back ground, but this leaves the reader little information on the question why this study was conducted. The outcome variables are characteristics of technologies that were mentioned in a reports from previous literature review. They are all process measures (country, clinical area, timeframe) but no outcomes of the forecasts are given. The results are listings of frequencies in the typology that the authors used. Introduction The introduction gives the reader little information on the reasons why this study is conducted. It argues that "To get the maximum benefit from forecasting, it is important that the methods used to identify emerging innovations and technologies are appropriate for the aims and time frame of the forecasting exercise and also as accurate as possible." Everybody would agree with this statement, and subsequently, I expected that the paper would provide me with information of the appropriateness and accurateness of the included studies or topics. However, the paper limits itself to a listing of characteristics of the topics that were found, and provide no confrontation with other data, leaving the question of all the topics were studied in an accurate manner or not. The intriguing question in the title is not answered, since the reader does not learn any qualitative or quantitative dimensions of the predictions. Again, this leaves me questioning what the value of the information of this paper is. Methods The authors build forth on their previous literature study, published in BMJ open in 2016. In that publication, the methods are described clearly, and it appears that the literature search is in order. The method section in this paper is rather limited and I struggled to find how the step was made from the 15 studies identified to the selection of topics that eventually result in the list of 896 topics that then becomes the data of interest. Was this done by text analysis? Were the 15 reports read and all topics marked and put on a list? Was this done manually, or with the search function of the text software? And if the 15 reports formed the basis of the topic list, why would this be a representative selection of topics that are of interest to the health community? The subsequent method presents the most frequent characteristics of the topics that were identified. I was wondering whether it would be possible to dig a bit deeper into the data and create cross tables to identify whether maybe policymakers were working on other
--

	topics than do physicians or a difference between government agencies and commercial organisations. This could maybe provide some information of a sense of combined urgency or a field of stakeholders all pursuing their own goals without much coherence in activities. The oldest studies in the literature are 45, 30 and 28 years old. I would expect that some of the forecasts made in those studies can be compared with empirical data in 2017. If this is added to the paper, it would yield information about the sensibility of these studies and may provide information on the accuracy of these types of forecasts. That would have implications for development of the field (maybe some tools work better than others) and implications for financiers of these studies. Of the studies prove to be accurate, it would make sense to keep doing them. But if you can show that they are not, it could reduce this part of health budgets to be spent on more beneficial work. The results are presented on clusters of the technology characteristics. The authors do not appear to have an intention to aggregate findings in order to be able to say anything about general directions or intentions of the process they study. Discussion The discussion stays rather close to the results of the analysis. I think that this inventory can be interpreted as an overview of the activities of the various stakeholders in the health community. Maybe when in the introduction more is said about the purpose and added value of this inventory, this provides opportunities to “zoom out” and provide more discussion about the meaning of the results to society or particular stakeholders that are involved in the process of innovation in health care. I believe that the data at hand can yield very interesting results, but that these results are interesting to a limited group of people. If the authors can provide more in-depth analyses and provide the reader with more information about the context, this is a potentially interesting study.
--	--

VERSION 1 – AUTHOR RESPONSE

Reviewer 1:

1. The article is interesting for policy makers as well as horizon scanners “in the weeds” doing the daily work of horizon scanning, though I suspect most interesting to the latter.

Thanks for your comments, and we believe that identifying and classifying the outputs of forecasting studies has value to health service planners, improving health system preparedness.

2. It illustrates some of the differences that emerge from horizon scanning done for different purposes and by different kinds of entities (e.g., commercial research, policy, governmental), though I believe readers would benefit from additional discussion of the differences. One naturally suspects that a great deal of commercial horizon scanning is not represented in the data sources, which may be worth acknowledging.

We agree and have noted this in a new sentence in paragraph 5 of the discussion.

3. I believe the introduction could include a sentence stating that some technologies are also used to enhance function, such as cosmetic enhancements, rather than just to prevent, diagnose or treat patients.

We agree and have added a sentence to the first paragraph of the introduction.

4. The data analysis provides an interesting tool to contrast and compare to the national horizon scanning activities conducted in the U.S. from 2010 through 2016 and those conducted in the EU. Looking at this analysis and U.S.-based analyses of the AHRQ horizon scanning activity previously published to the AHRQ website, one can discern both similarities and differences in priority setting, timeframes used for scanning and forecasting, target clinical areas of interest, and technology types. The AHRQ system was halted December 31, 2015, but until then was the largest system in the world, I believe. It may be worth a mention.

We agree that the AHRQ system was of great benefit and complemented the outputs of other horizon scanning agencies across the world. The outputs from AHRQ were the result of comprehensive and ongoing horizon scanning and priority setting activities, and as such they mirror similar systems elsewhere in the world, such as that of our own unit (NIHR Horizon Scanning Research & Intelligence Centre, UK) and other members of the EuroScan International Network (<https://www.euroscan.org/>). However, the activity and methods adopted by such organisations differs somewhat from the one-off or repeated forecasting exercises identified as part of our systematic review and (in many cases) published in the biomedical literature. As such, we have not made specific reference to AHRQ, or any other horizon scanning system.

5. The broad categories used for classifying technology type are less informative than more granularly defined categories. For example, there may be interesting trends to observe in the “drugs” category—such as “targeted therapies” or “biologics”—but that would require more granular analysis and categorization that may not be possible from the studies from which data were abstracted.

We agree that further classification would be useful, but as you mention, this wasn't feasible from the data available from the included studies.

Reviewer 2:

Abstract:

The objective is to describe and classify a number of health technologies predicted in forecasting studies. Maybe the format of the journal does not allow for presenting an back ground, but this leaves the reader little information on the question why this study was conducted.

We added an additional sentence to the last paragraph of the introduction section, but as you note, the format of the journal does not allow a background section in the abstract.

The outcome variables are characteristics of technologies that were mentioned in a reports from previous literature review. They are all process measures (country, clinical area, timeframe) but no outcomes of the forecasts are given.

The results are listings of frequencies in the typology that the authors used.

The published forecasts did not describe outcomes or predicted impacts of the technologies they mention, whether in terms of the number of patients, level of benefit to patients/community or likely level of diffusion and uptake. We have now made the purpose of the study clearer in the paper title, abstract and introduction so that the fact that this is a descriptive analysis derived from analysing previous forecasts identified in a previous systematic review is now explicit.

Introduction

The introduction gives the reader little information on the reasons why this study is conducted. It argues that “To get the maximum benefit from forecasting, it is important that the methods used to

identify emerging innovations and technologies are appropriate for the aims and time frame of the forecasting exercise and also as accurate as possible.” Everybody would agree with this statement, and subsequently, I expected that the paper would provide me with information of the appropriateness and accurateness of the included studies or topics. However, the paper limits itself to a listing of characteristics of the topics that were found, and provide no confrontation with other data, leaving the question of all the topics were studied in an accurate manner or not. The intriguing question in the title is not answered, since the reader does not learn any qualitative or quantitative dimensions of the predictions. Again, this leaves me questioning what the value of the information of this paper is.

We have amended the title of the manuscript and the abstract (as suggested by the editor) to more clearly describe the purpose of this paper. We have also added a sentence to the last paragraph of the introduction to ensure the reader is fully aware as to the purpose of this work (as described above). Further comments on the accuracy of forecasts are given in response to your comment below.

Methods

The authors build forth on their previous literature study, published in BMJ open in 2016. In that publication, the methods are described clearly, and it appears that the literature search is in order. The method section in this paper is rather limited and I struggled to find how the step was made from the 15 studies identified to the selection of topics that eventually result in the list of 896 topics that then becomes the data of interest. Was this done by text analysis? Were the 15 reports read and all topics marked and put on a list? Was this done manually, or with the search function of the text software?

The technologies were identified through careful reading of the published text and tables for each forecast. We have added a sentence to paragraph 3 of the methods to make this clearer.

And if the 15 reports formed the basis of the topic list, why would this be a representative selection of topics that are of interest to the health community?

The basic topic list is derived from all identified and available forecasts, as forecasting and the different methods employed by different groups is our main interest. Of course, we recognise that the outputs of these forecasting exercises may not be the technologies or fields of most interest to health professionals and they may not generate technologies that match health service priorities. In that sense, they are technology driven rather than needs driven. We have discussed the intended uses for different types of identified technology in our discussion section to allow the reader to judge the importance of these.

The subsequent method presents the most frequent characteristics of the topics that were identified. I was wondering whether it would be possible to dig a bit deeper into the data and create cross tables to identify whether maybe policymakers were working on other topics than do physicians or a difference between government agencies and commercial organisations. This could maybe provide some information of a sense of combined urgency or a field of stakeholders all pursuing their own goals without much coherence in activities.

We agree and have added the extra table as suggested. We did partially deal with these comments in the original text of the Results section, noting that “Regenerative medicine approaches were the most commonly forecast technologies (23.2%) by studies with a commercial intent while diagnostic tests and imaging were the most commonly forecast technologies by studies with policy planning and research intentions (20.6% and 23.4% respectively) ($p < 0.001$)”. However, we have followed this with a further comment in both the Results – Technology type and Results – Technology purpose sections, building on the table mentioned above.

The oldest studies in the literature are 45, 30 and 28 years old. I would expect that some of the forecasts made in those studies can be compared with empirical data in 2017. If this is added to the paper, it would yield information about the sensibility of these studies and may provide information on the accuracy of these types of forecasts. That would have implications for development of the field (maybe some tools work better than others) and implications for financiers of these studies. Of the studies prove to be accurate, it would make sense to keep doing them. But if you can show that they are not, it could reduce this part of health budgets to be spent on more beneficial work.

See comments to editor above. We agree that an assessment of the accuracy of some of the forecasts made would be feasible at this point, but this work is beyond the scope of this project at present and is not something we can address at this time. We believe a descriptive analysis of the output of forecasts, including a classification of the types of technology predicted by different organisations and processes, is still of value to health service planners, commissioners of research and horizon scanners.

The results are presented on clusters of the technology characteristics. The authors do not appear to have an intention to aggregate findings in order to be able to say anything about general directions or intentions of the process they study.

Please see our response to earlier comments.

Discussion

The discussion stays rather close to the results of the analysis. I think that this inventory can be interpreted as an overview of the activities of the various stakeholders in the health community. Maybe when in the introduction more is said about the purpose and added value of this inventory, this provides opportunities to “zoom out” and provide more discussion about the meaning of the results to society or particular stakeholders that are involved in the process of innovation in health care. I believe that the data at hand can yield very interesting results, but that these results are interesting to a limited group of people. If the authors can provide more in-depth analyses and provide the reader with more information about the context, this is a potentially interesting study.

As described above, we have added an additional table and comments in the results section to increase the depth of our reporting. We have added a sentence to both paragraphs 2 and 3 of the Discussion section to reflect on the results from newly added table.

VERSION 2 – REVIEW

REVIEWER	Jeffrey Lerner ECRI Institute, USA, UK, Malaysia ECRI Institute, a nonprofit organization, carries out horizon scanning.
REVIEW RETURNED	10-Apr-2017

GENERAL COMMENTS	I think your article will be useful to forecasting professionals and those supporting them such as research librarians. I believe you would need to expand the explanation of why policymakers will value the analysis for them to do so. At the moment you state they
--

	should be interested, but this seems to rest on the assertion that forecasting itself would help them make better decisions. It is logical that the latter is true, but since both a more granular analysis and a way of evaluating outcomes are beyond the scope of your paper, I think they won't be satisfied. Perhaps you could add language that links to research that you think does address these latter issues. I checked the box for major revision because I think the issues are important to address, not because this is difficult to accomplish.
--	---

VERSION 2 – AUTHOR RESPONSE

Thank you for your further comments. We absolutely agree that additional detail and examples would help readers understand how to use forecasting information to improve planning and decision making, however the literature in this area (how forecasting intelligence has been used by policy makers in the health sphere) are sparse, and there are no key examples at the system level that can easily be quoted from the published literature. Therefore, we have sought to strengthen the first paragraph of the discussion section to make clear that the value of such intelligence lies in having systems in place to receive act on it, and drawing attention to previous work done in the early awareness field (shorter-term predictions), where published examples do exist.

VERSION 3 – REVIEW

REVIEWER	Jeffrey Lerner ECRI Institute, USA, UK, Malaysia, Dubai
	ECRI Institute carries out forecasting
REVIEW RETURNED	28-May-2017

GENERAL COMMENTS	Although I have recommended acceptance, I do have two suggestions relating to the abstract and one comment on the discussion. I believe that under Main Outcome Measure in the abstract (p.3 line 25) you should consider replacing "Outcomes related to..." with "A portrait describing...". In the conclusion in the abstract you might consider reversing the content of the sentences and combining into one sentence such as "While further research...description and classification..." My comment is that that the Discussion section of the article is very thoughtful.
--